# Continuous fluorescence-based monitoring of seawater pH *in situ*.

John W. Runcie[1,2,3], Christian Krause[4], Sergio A. Torres Gabarda[1,3]. Maria Byrne[1,3]

[1]School of Life and Environmental Sciences, University of Sydney, Sydney, Australia

[2]Aquation Pty Ltd, Umina Beach, Australia,

[3]Sydney Institute of Marine Sciences, Mosman, Australia

[4]Presens Precision Sensing GmbH, Regensburg, Germany

*Correspondence to*: John W. Runcie (john.runcie@sydney.edu.au)

**Abstract.** Electrical conductivity (salinity), temperature and fluorescence-based measurements of pH were employed to examine diel fluctuations in seawater carbonate chemistry of surface waters in Sydney Harbour over two multiple-day periods. A proof of concept device employing the fluorescence-based technique provided a useful time-series for pH. Alkalinity with pH and temperature were used to calculate the degree of calcite and aragonite saturation ($\Omega_{Ca}$ and $\Omega_{Ar}$ respectively). Alkalinity was determined from a published alkalinity-salinity relationship. The fluctuations observed in pH over intervals of minutes to hours could be distinguished from background noise. While the stated phase angle resolution of the lifetimes fluorometer translated into pH units was $\pm 0.0028$ pH units, the repeatability standard deviation of calculated pH was 0.007 to 0.009. Diel variability in pH, $\Omega_{Ar}$ and $\Omega_{Ca}$ showed a clear pattern that appeared to correlate with both salinity and temperature. Drift due to photodegradation of the fluorophore was minimised by reducing exposure to ambient light. $\Omega_{Ca}$ and $\Omega_{Ar}$ fluctuated on a daily cycle. The net result of changes in pH, salinity and temperature combined to influence seawater carbonate chemistry. The fluorescence-based pH monitoring technique is simple, provides good resolution and is unaffected by moving parts or leaching of solutions over time. The use of optics is pressure insensitive, making this approach to ocean acidification monitoring well suited to deepwater applications.

## 1 Introduction

Ocean carbon chemistry is predicted to vary in response to elevated concentration of atmospheric carbon dioxide ($CO_2$), with a decline in pH and an increase in the partial pressure of dissolved $CO_2$ ($pCO_2$) over the coming decades. However, short-term diel changes in seawater pH and $pCO_2$ at the scale of an embayment are unlikely to represent predicted globally increasing trends. Seawater carbonate chemistry in shallow nearshore environments is more likely driven by a combination of biological (photosynthesis, respiration) and local hydrodynamic (tidal, low salinity surface- and/or ground-water input) processes (Santos et al. 2012) with typical variation at time scales ranging from minutes to days. Capturing these short-term fluctuations requires virtually-continuous monitoring of water chemistry.

Seawater pH has been commonly measured with a potentiometric technique using glass electrodes. While this method can be accurate and precise if appropriate care is taken with temperature control and the use of seawater buffers (Dickson, 2010), this technology suffers from gradual drift that may be attributable to changes in the strain-induced asymmetry potential of

the glass bulb and reference junction effects. In addition, common reference electrode designs are susceptible to poisoning, in particular by sulphide released when conditions become anoxic. However, recent advances in electrode design including the incorporation of double (or even quadruple) junction salt bridges have enabled the continuous use of glass pH electrodes in shallow marine waters for over 12 months (Ionode Pty Ltd, Australia, pers. comm).

Spectroscopic techniques for measuring seawater pH use a pH-sensitive dye that assumes different absorbance spectra depending on pH. While the method can provide pH estimates with an uncertainty less than 0.01, variability in the purity of the dye obtained from commercial suppliers can cause the apparent extinction coefficients associated with a particular dye to differ slightly from published extinction coefficient values. Consequently, uncertainty associated with this technique is generally assumed to be about 0.015 pH units (Douglas and Byrne 2017b).

Ion selective field effect transistor (ISFET) technology is becoming more widely used to measure pH in seawater. pH sensors using ISFET technology comprise the ISFET sensor itself - a solid-state device with a H+ ion sensitive material, a counter electrode and a reference electrode. In pH monitoring systems such as the SeaFET (Martz et al. 2010, Takeshita et al. 2015, Johnson et al, 2016) a chloride electrode is used as reference as chloride ion activity is relatively constant in deep water. However this introduces problems in locations where chloride concentrations may rapidly fluctuate, such as some euryhaline estuarine environments.

Optical fluorescence-based approaches to pH sensing offer an alternative to glass electrodes, spectrophotometric techniques and ISFET sensors. Optical fluorescence-based methods do not require electrolyte solutions or gels which gradually lose ions, nor do they require reference electrodes and their accompanying susceptibility to varying environmental conditions. Optical methods are also impervious to high pressure. Only the housing for the device and the window through which the optical signal is passed must be pressure resistant. The use of fluorescence lifetime measurements provides the additional advantage of being insensitive to changes in fluorescence intensity. As the fluorophores used in this study are sensitive to changes in temperature and salinity, careful measurement of these parameters remains essential.

In this study we explore a new approach to logging pH in seawater in the setting of an ocean-water dominated estuarine environment. We incorporated a commercially available fluorescence-based pH sensing system (Presens GmbH) in a submersible continuous monitoring device (Aquation Pty Ltd) to measure short-term changes in seawater pH in surface waters in Sydney Harbour over two multi-day intervals in austral Autumn and Summer. Electrical conductivity and temperature were also measured, and an approximation for alkalinity was derived from a salinity-alkalinity relationship reported for the Australasian region (Lenton et al. 2016). We acknowledge that the relationship between salinity and alkalinity is subject to terrestrial influence, and consequently our estimate of alkalinity remains an approximation. With these data we estimated changes in seawater carbonate chemistry during the deployments. The objectives of the study were a) to characterise the fluorescence-based pH monitoring system in seawater in terms of stability and resolution, and b) to characterise diel fluctuations of pH and aragonite/calcite saturation state in surface waters. The study focussed on natural

fluctuations in seawater pH and the capability of the fluorescence technique to observe these fluctuations. Determining the uncertainty of pH values obtained from the fluorescence device was not an objective, as precise and accurate pH determinations would require considerable resources that were not available at the time of study.

## 2 Methods

### 2.1 Study site and sampling arrangements

Real-time measurements were made of seawater in the Sydney Harbour estuary at the Sydney Institute of Marine Sciences (SIMS), Chowder Bay, NSW, Australia (-33.839357 °S, 151.254587 °E) (Fig. 1). Chowder Bay is a shallow (5 to 10 m) embayment located on the northern shore of Sydney Harbour some 3 km from the ocean. Tides in Sydney Harbour (depths typically ~20 m) are semi-diurnal, with a maximum difference of 1.95 m at the estuary entrance. Tidal fluctuation during the deployment intervals is referenced to measurements made at nearby Fort Denison (Caldwell et al. 2015). Seawater chemistry was measured from ~1 m below sea level at low tide below the SIMS pier at Chowder Bay.

Two field deployments were conducted. Firstly, water was collected ~1 m below low tide and pumped 3-5 m up into a 2 L reservoir in an enclosed structure at the end of the pier. Water exited the reservoir through a narrow tube located below an overflow pipe at the upper part of the reservoir, passing through a clear polycarbonate tube for optical measurement (*ex situ* measurements, Autumn, 8th-19th April 2016). Fluorescence spots were attached to the inside of this tube and a plastic optical fibre was positioned perpendicular to the outside of the tube and directed to the spot. To avoid photobleaching the system was covered in black plastic eliminating most light. Waste water was directed back into the harbour at least 5 m away from the intake location. This preliminary series of tests enabled us to make frequent measurements as the system could use a constant supply of mains power. The primary objective of the *ex situ* measurements was to assess the stability of the system.

*In situ* measurements were later made ~1 m below low tide during 14th-22nd December 2016, with the self-contained instrument (Fig. 2) attached vertically to a post under the pier and the fluorescence spot facing downwards to minimise sunlight-induced photobleaching. In this arrangement the plastic optical fibre within the clear polycarbonate cylindrical housing of the instrument was oriented perpendicular to the faceplate. To ensure that the spot could not peel off, a non-fluorescent mesh was held in place over the spot with a plastic ring. This arrangement ensured that the spot was fixed in position and water could easily reach the fluorophore and respond to ambient pH. A small hole was drilled partway into the inside of the faceplate to bring the fibre tip closer to the spot, which was affixed to the external surface of the faceplate. This was necessary to increase signal strength to an acceptable level. During the *in situ* measurements, water samples were collected adjacent to the instrument and removed to the laboratory for measurement of $pH_T$ (see below) within 30 minutes of sampling. The instrument contained three NiMH battery packs in series comprising 14 cells each with a total of 13.5 Ah at 16.8 volts.

Sunrise and sunset at the study site on the 9[th] April 2016 was 06:14 and 17:38 respectively. Sunrise and sunset on the 18[th] December 2016 was 05:40 and 20:02 respectively.

## 2.2 pH measurements

The fluorescence-based pH measurement uses a dual-luminophore frequency-domain fluorescence decay technique (Huber et al. 2001). A pH sensitive dye with short fluorescence decay time which changes its fluorescence intensity dramatically with pH is mixed together with an inert reference dye with long decay time. Both dyes are immobilized in particles which are dispersed in a hydrophilic membrane. This sensing membrane is in contact with the ambient water and a light guide is brought to the opposite site of the membrane to read the pH (Fig. 3). The read out unit (EOM-pH-mini) sends sine-wave modulated blue light to excite both dyes simultaneously (the "spot") and measures the average decay time of the sum of returned fluorescence light from both dyes.

Calibration was performed prior to deployment of the instrument. All glassware, buffers, seawater and the instrument were left overnight in an air-conditioned room to reach a stable temperature of 22 °C. A pH cell with an Ag/AgCl reference electrode (Ionode Pty Ltd) was calibrated against standard buffers (pH 7.02 and 10.06 at 20°C, Amalgamated Instruments Co. Pty Ltd, Australia). A pair of phosphate buffer solutions were also prepared, isotonic to ~34.5 ppt NaCl ($I = 0.7$ M) with B1=8.33 pH and B2=4.00 pH (where B1 and B2 are the measured pH of each of the buffers). A small volume (e.g. 0.25 mL) of B1 was added to 10 mL of B2 until the combined buffer reach a pH of 4.45 as measured by the pH cell. An aliquot (approx. 1 mL) of this combined buffer solution of known pH was pipetted onto the fluorescence spot (fixed to the inside of the clear polycarbonate tube, or fixed to the faceplate of the instrument) and the fluorescence phase measured. Settling time of the phase value for each aliquot could take at least 10 minutes. Then, more B1 was added to the combined buffer and the process repeated at pH 5.66, 6.81, 7.23, 7.72, 8.02 and 8.30. The relationship between measured phase and (total) pH was determined with a non-linear Boltzman sigmoid model using the software provided by the manufacturer. We report pH in the total hydrogen ion scale (as recommended by Dickson 2010). Both room and buffer temperature were stable during the calibration procedure at 22°C and 22 ± 0.5 °C, respectively.

At the beginning and end of the *in situ* deployment, water samples were collected adjacent to the submerged pH monitor and transported to the nearby lab at SIMS. At each sampling time, three 200 mL glass jars were completely filled, and lids secured underwater to eliminate air bubbles. This process was repeated at least three times, with approximately 30 minutes between each sampling time. Within 30 minutes of collection, pH was measured spectrophotometrically with *m*-cresol purple indicator dye (Acros Organics, Lot # A0321770) using a custom-built automated system with a USB4000 fibre optic spectrophotometer (Ocean Optics) following SOP 6b from Dickson et al. (2007) and calculations from Liu et al. (2011). Spectrophotometric measurements of $pH_T$ were frequently validated using seawater certified reference material (CRM, Dickson Standard Batch 145). The repeatability of the spectrophotometric technique in our laboratory is typically 0.006 at

temperatures between 17 to 24°C (standard deviation), with an accuracy very much dependent on the dye quality. We did not apply corrections for dye impurities, temperature and salinity variations as discussed in Douglas and Byrne (2017a,b) because obtaining such highly accurate  pH values was not an objective of this study. The uncertainty attributable to these combined factors is within approximately 0.015 (Douglas and Byrne (2017b). The measured $pH_T$ was corrected according to the calculated $pH_T$ of the CRM using published values for total alkalinity ($A_T$), dissolved inorganic carbon (DIC) and salinity at the same temperature.  This correction eliminates much of the variability due to dye quality, light source and cuvette scratches.

A subset of consecutive pH measurements over a 100 minute period was sampled during each of the two deployments to determine the variability in measured pH as repeatability (standard deviation).  The resolution attributable to the lifetime decay fluorometer (as stated by the manufacturer) can be matched with corresponding values of pH to provide resolution of pH measurement at the instrument level.  Corrections for the *in situ* measurements were made against spectrophotometric measurements of the reference samples by calculating two average offsets between values of collected samples measured using the fluorometric device and the spectrophotometric device.  One offset was derived for the beginning of the deployment and one for the end.  The gradual decline in measured values over time was corrected assuming a constant linear drift.

*In situ* measurements were taken ten times per minute and averaged over one minute before analysis.  *Ex situ* measurements were taken once per minute and a running average of 10 minutes of the *ex situ* measurements is presented.

## 2.3 Temperature

PT100 sensors were either suspended in the 2L reservoir that received seawater immediately prior to *ex situ* measurements, or were simply positioned up against the inside of the housing wall when deployed *in situ*.  Initial temperature data during the *in situ* deployment were ignored until the sensors reached equilibrium with ambient water.  *In situ* measurements were taken ten times per minute and averaged over one minute before analysis.  *Ex situ* measurements were taken twice per minute and a running average of 10 minutes of the *ex situ* measurements is presented.

## 2.4 Salinity

Electrical conductivity was measured using a Hanna HI3001 four ring potentiometric conductivity probe designed for flow-through sensing wired to a custom circuit board.  Millivolt values were converted to salinity assuming a linear relationship between sensor readings and salt content.  The sensor was calibrated directly against a pure NaCl solution across a range of concentrations (0 to 40 ppt) at temperatures similar to that expected of seawater at the sampling location (23°C).  Uncertainty due to differences in conductivity of a pure NaCl solution at 35 ppt and S=35 was assumed to be small.  *In situ*

measurements were taken ten times per minute and averaged over one minute before analysis. *Ex situ* measurements were taken three times per minute and a running average of 10 minutes of the *ex situ* measurements is presented.

## 2.5 Datalogging and calculations

Both flow-through and *in situ* deployments were conducted using a custom datalogger/controller system based on a Submersible Datalogger (Aquation Pty Ltd, Australia). All data were saved on board and downloaded to PC after retrieval of the instrument. Alkalinity was calculated from salinity data according to a linear salinity-alkalinity relationship for Australasian waters described by Lenton et al. (2016). Drift and offset corrections were made to the raw data as required (described above), and final pH, salinity, temperature and alkalinity values were used (with equilibrium constants from Mehrbach et al., 1973 refit by Dickson and Millero, 1987) to calculate $\Omega_{Ca}$ and $\Omega_{Ar}$ in CO2Sys (V2.1) (Dickson and Millero 1987). The final pH is reported on the total scale.

## 3.0 Results

### 3.1 pH

Seawater collected and pumped to the sensor varied by 0.08 pH units from 7.97 to 8.05 over the three-day interval in Autumn (Fig. 4). The phase angle resolution of the lifetimes fluorometer was stated by the manufacturer as 0.05 degres, and this translated to an instrument resolution of 0.0028 pH units when calibrated against the phosphate buffer mixtures. The measurement technique is also constrained by the stated resolution of 0.01 pH units) of the fluorescence spots. The repeatability standard deviation of calculated $pH_T$ (derived from 100 minutes of consecutive measurements during each deployment was 0.009 and 0.007 pH units for the April and December deployments respectively. The best precision can be achieved by averaging multiple measurements, although this will of course hide real variability. Gradual bleaching is known to influence fluorophores (Lakowicz 2006), contributing to a steady drift. This was accounted for with corrections based on samples collected *in situ* as described above. In this case, the decline in pH was 0.0136 pH units per day which is approximately three-fold the drift associated with measurement-induced bleaching of 0.003 pH units per 1000 measuring points as stated by the manufacturer. pH measured over summer ranged from 7.95 to almost 8.15 units, some 0.1 units more basic than Autumn values (Fig. 5).

### 3.2 Temperature

Surface seawater temperature fluctuated between 21.6°C and 22.6°C in April (*ex situ* measurements, Fig. 4), and 21.0 to 23.6 in December (*in situ* measurements, Fig. 5). The stated resolution of the PT100 temperature sensors was 0.01°C and

concurrent readings were within 0.05 °C of each other.  Over the three day period, temperature showed a consistent pattern of declining at night and then rapidly increasing again around the middle of the day, reaching a maximum value around mid-afternoon. During the *in situ* deployment, diel fluctuations in ambient seawater temperature varied between 1 and 2°C, and increased to 23.6°C in the last two days of the measuring interval (Fig. 5).

**3.3 Salinity**

Salinity was calculated from electrical conductivity, and ranged from 33.0 to 34.3 ppt during the *ex situ* measurements, and 34.6 to 37.7 during the *in situ* measurements with a mean value of 36.5 ppt during the *in situ* measurements (Figs 4 and 5 respectively).  Diel variation tracked the tidal cycle with higher salinity driven by the incoming tide (Fig. 4).  However, the amplitude of the salinity signal was not consistently coincident with the tidal amplitude (Fig. 5).

**3.4 Saturation states**

Both $\Omega_{Ca}$ and $\Omega_{Ar}$  calculated from water in the *ex situ* trial fluctuated over the course of three days, providing a smoothed approximation of diel variation in these two parameters (Fig. 6).  Most striking is the noise associated with the calculated values, reaching as high as 1 $\Omega$ units.  As a consequence of the definition of saturation state and due to the higher solubility of aragonite, and from specific values of $K_{sp}$ for aragonite and calcite, $\Omega_{Ar}$ was slightly less variable than $\Omega_{Ca}$ (with values

between 2.5 and 3.1, or 3.8 to 5.4 units respectively, Fig. 6).

**4.0 Discussion**

**4.1 Instrumentation**

The fluorescence-based pH measurements provide sufficient resolution to observe and predict diel changes in nearshore system pH and indicated significant changes in seawater /carbonate chemistry at timescales of hours to days.  Values of pH,

salinity and temperature were all within what one would expect for the harbour at that time of year (e.g. pH ~8.0, S=~35 ppt, T= ~18°C, pers. obs.).  Further, the average magnitude of both $\Omega_{Ca}$ and $\Omega_{Ar}$ were within the bounds of expected values for current-day marine surface waters.  While an attempt was made to minimise exposure of the sensor spot to ambient light, the rate of drift observed in the instrument due to photobleaching may have been greater during the day.  Photobleaching induced drift could be further reduced by excluding all ambient light and by reducing the sampling frequency.  This would

be needed for longer term deployments.  Calibration against high accuracy pH measurements at least at the beginning and end of a deployment, and more frequently for an extended deployment, would be required to maintain suitably accurate data.

The mains-powered instrument designed to measure water pumped up from the harbour had the advantage of unlimited power and the capacity to make multiple measurements several times per minute. However, the dissolved gas content of pumped water can vary relative to the source water due to degassing and temperature changes, making measurements

potentially less representative of the water to be sampled. We kept this to a minimum by pumping water over a short distance. Further, the need to use a suitable secure location with a constant power source is a limitation with respect to field deployments. In contrast, the self-powered instrument tested during December 2016 had sufficient power to operate unattended for several weeks, and with a decrease in sampling frequency the lifetime of the unit could be extended to several months. For even longer periods the device could be constructed using a longer cylinder, a separate battery pack, with

lithium ion batteries, or combinations of these. With this approach, the unit could be powered to operate unattended for over 12 months.

The housing used in this study was constructed from 5 mm thick acrylic. However, calculations indicate failure of the housing at depths exceeding 100 m. A metal housing with regular bulkheads along its length would increase pressure resistance and provide an instrument capable of withstanding deep ocean pressures, although a system with improved

precision would be required for deep ocean work as the variation in pH at depth is slight relative to surface waters.

### 4.2 Data and significance

While there was a clear diel fluctuation in temperature of about 1 ℃ both in December and April, the gradual increase in temperature during the 20$^{th}$ and 21$^{st}$ of December (austral Summer) was coincident with a decline in salinity and tidal amplitude (Fig. 5). This suggests an influx of warmer freshwater into the estuary coupled with weaker tidal pumping, where

pumping corresponds with flushing of the harbour with higher-salinity seawater from the ocean. pH, $\Omega_{Ca}$ and $\Omega_{Ar}$ also declined during these last two days of the deployment. These data show the important role that temperature and salinity play in determining seawater carbonate chemistry (Millero 1995). Further, as the saturation state in this study is in part derived from alkalinity which is in turn derived from salinity and temperature, the importance of obtaining accurate measurements of temperature and salinity is to be emphasised. As a consequence of these physical factors (and presumably photosynthesis

and respiration; Schulz and Riebesell, 2013), nearshore coastal waters and estuaries experience changes in $\Omega_{Ca}$ and $\Omega_{Ar}$ not only over the course of several days, but also over a 24 hour period. This contrasts with oceanic marine waters that maintain relatively constant values of saturation state. Models used to forecast seawater carbonate chemistry, such as those employed in the IPCC reports are typically based on the open ocean. The projections for the open ocean do not incorporate the short-term fluctuations typical of coastal waters (Hendriks et al. 2015).

When assessing nearshore shallow-water marine coastal and estuarine environments, these short-term fluctuations in seawater carbonate chemistry can be significant, as found in this study, and are best accounted for and included in predictive models. Perhaps most important is to determine the capacity for organisms that live in these highly fluctuating environments

to withstand or thrive in spite of such a regular chemistry change. In this study we generated a targeted data set to assess the application of the fluorescence-based method for measuring seawater pH. Our results show the influence of fluctuating temperature and salinity on calcium carbonate saturation state in an estuarine environment. More research is needed to determine in detail these and other drivers of fluctuating saturation states (e.g. photosynthesis and respiration), and the actual consequences of these changes to key resident marine species.

### 4.3 Conclusion

We observed pH of surface water at Chowder Bay in Sydney Harbour to vary significantly across a diel cycle, influenced by temperature, salinity and tidal height /alternating freshwater or saline dominance. The fluorescence-based pH monitoring technique used here could resolve changes in pH to 0.007 pH units. Corresponding saturations states of calcite and aragonite ($\Omega_{Ca}$ and $\Omega_{Ar}$), calculated from pH, temperature and salinity-derived alkalinity values showed diel fluctuations in seawater carbonate chemistry that can be interpreted as the combined effect of pH, temperature and salinity. Variability in saturation state over such short time scales in this shallow ecosystem indicates the importance of high temporal and spatial resolution measurements of seawater carbonate chemistry when determining the impact of ocean acidification on shallow nearshore marine and estuarine ecosystems.

### Competing Interests

JWR is a director of Aquation Pty Ltd.

### Acknowledgements

This paper is contribution number 209 from the Sydney Institute of Marine Science.

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

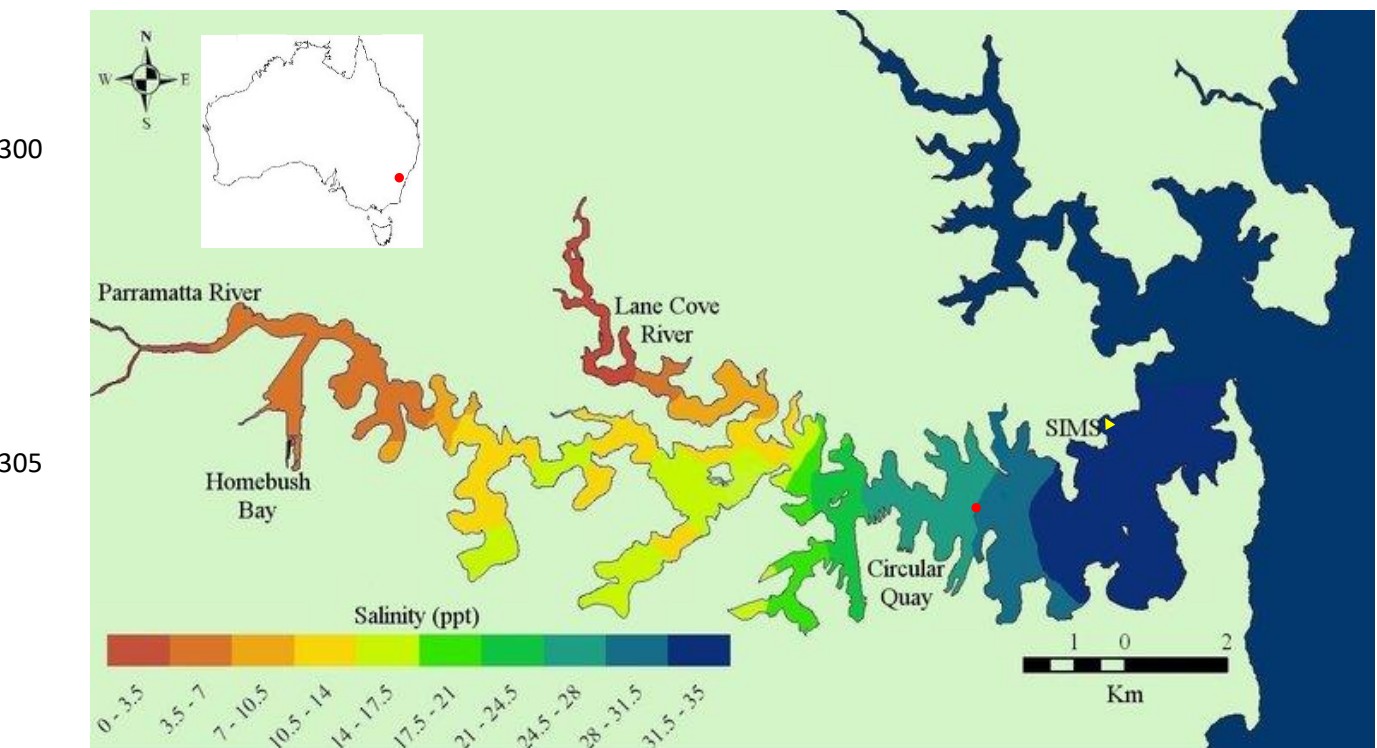

Parramatta River

Lane Cove
River

SIMS

Homebush
Bay


Circular
Quay

Salinity (ppt)

1  0  2

Km

0 - 3.5   3.5 - 7   7 - 10.5   10.5 - 14   14 - 17.5   17.5 - 21   21 - 24.5   24.5 - 28   28 - 31.5   31.5 - 35

**Figure 1. Port Jackson (Sydney Harbour) with sampling site (SIMS, yellow triangle) and Fort Denison tide reference station (red circle). The red dot on the insert map indicates Sydney's location. Map courtesy of http://www.sydneyharbourobservatory.org/.**

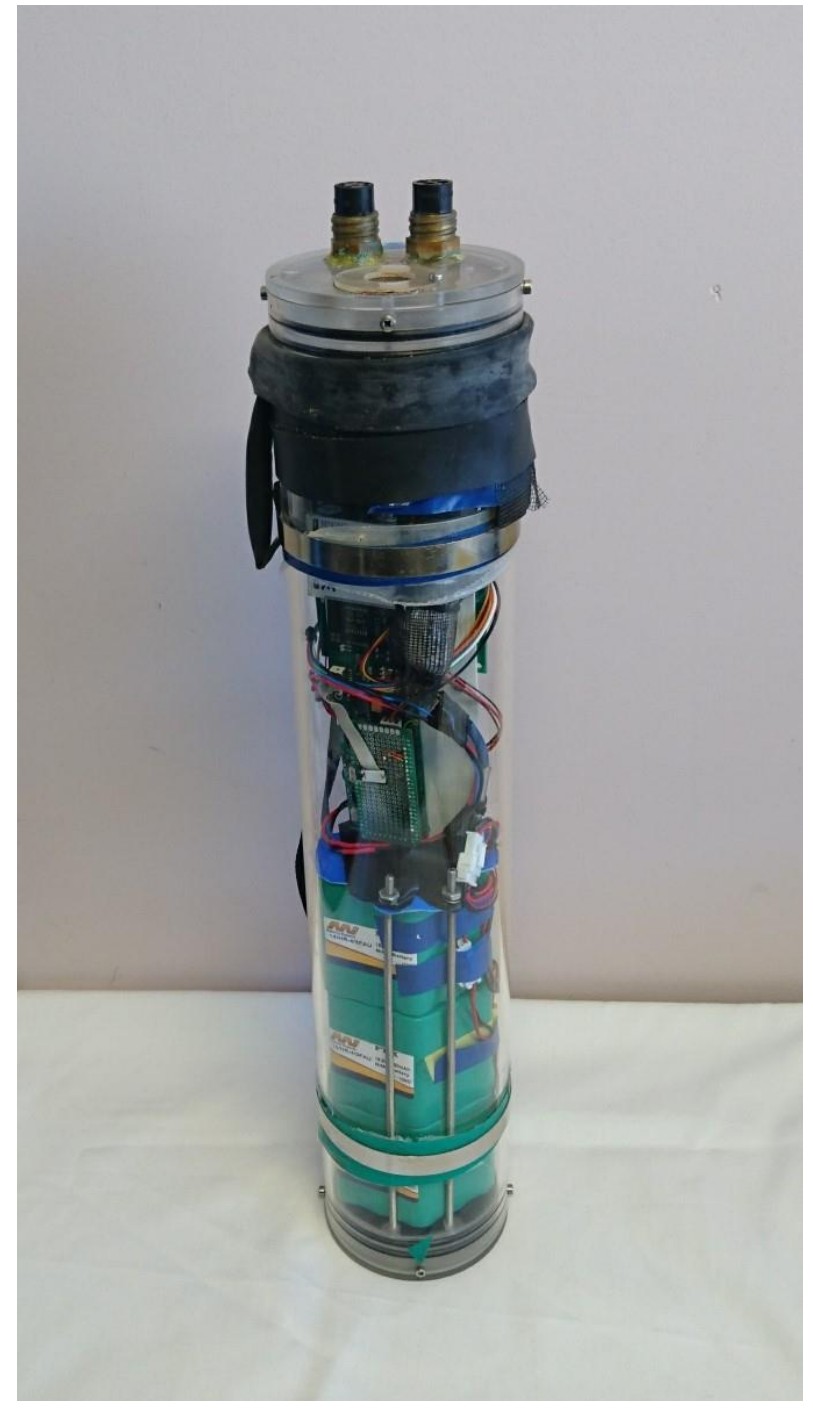

**Figure 2. Submersible logging pH monitor. The fluorescence spot is located on the upper faceplate between and forward of the two submersible connectors, beneath a clear plastic ring.**

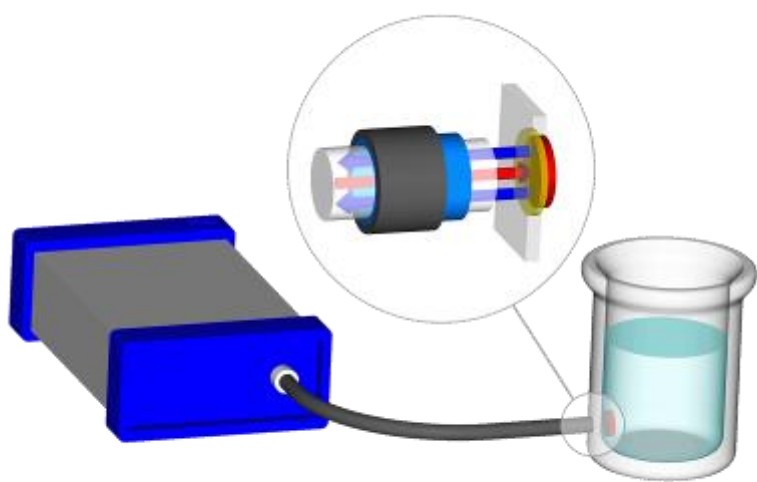

**Figure 3. Schematic of pH monitoring system with EOM-pH-mini (left), and fibre optic conveying light to fluorescent spot attached to inside of vessel (right). The insert shows light conveyed through the transparent vessel wall to the spot (red arrow), and fluorescent light emitted from the spot back along the fibre (blue arrows).**


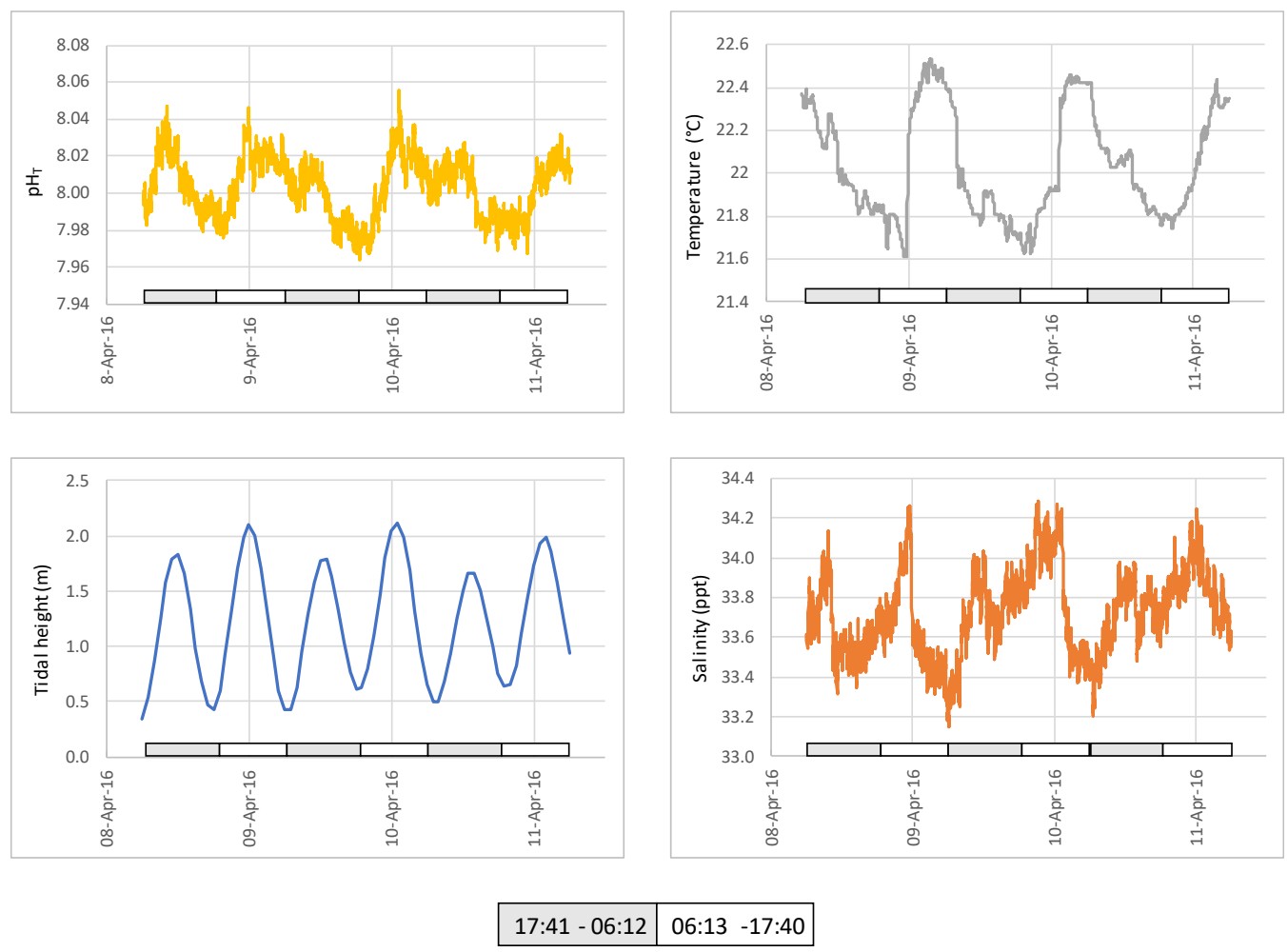


Figure 4. pH, temperature, tidal height and salinity of seawater pumped from ~1 m below low water at Chowder Bay, Sydney over a three-day period. Grey and white boxes indicate the time between sunset and sunrise.


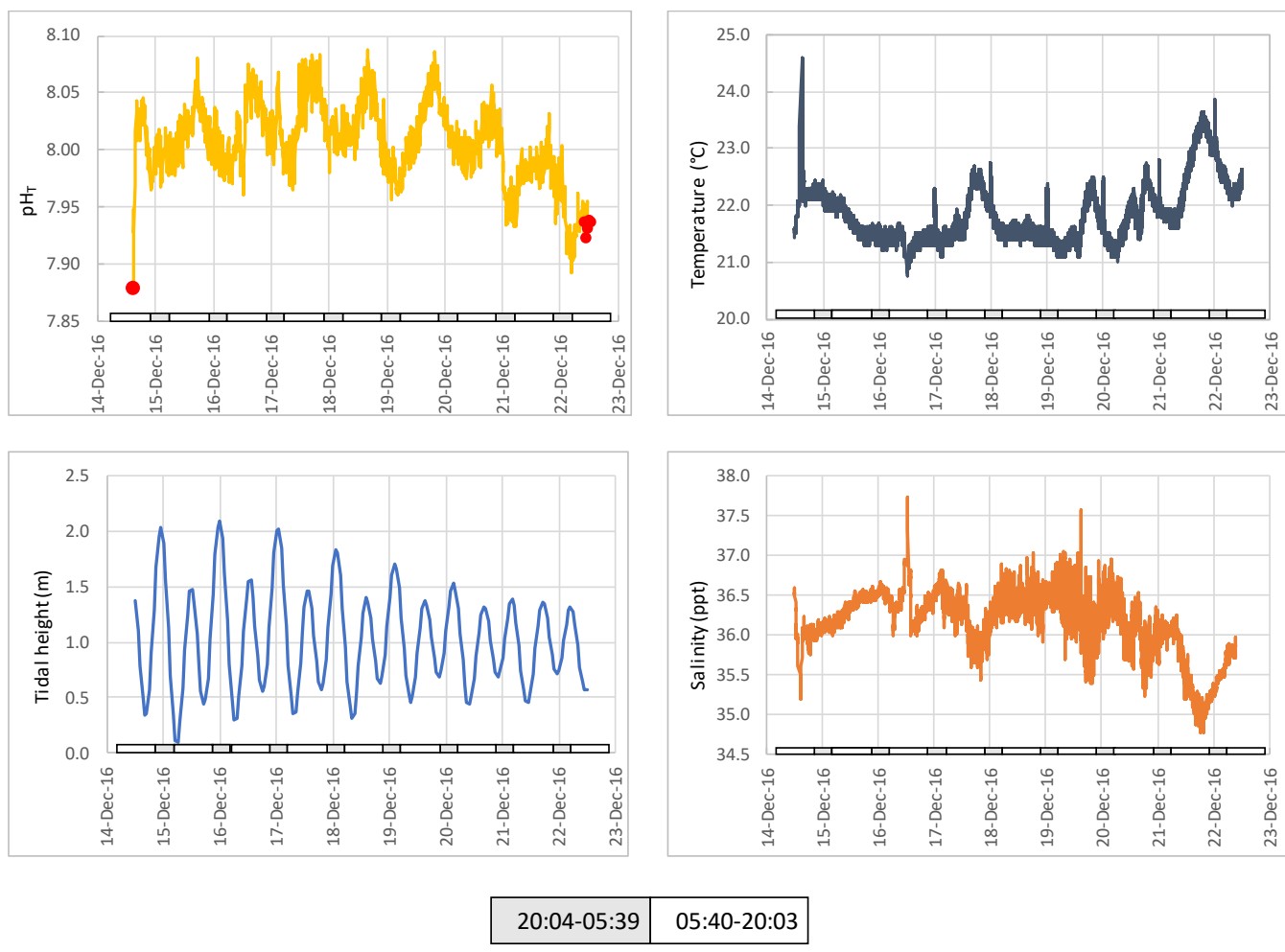

20:04-05:39 | 05:40-20:03

**Figure 5. pH, temperature, tidal height and salinity of seawater measured *in situ* at ~1 m below low water at Chowder Bay, Sydney over a multi-day period. Grey and white boxes indicate the time between sunset and sunrise. Red dots indicate values of reference**
**samples collected adjacent to the device and measured using spectrophotometry.**

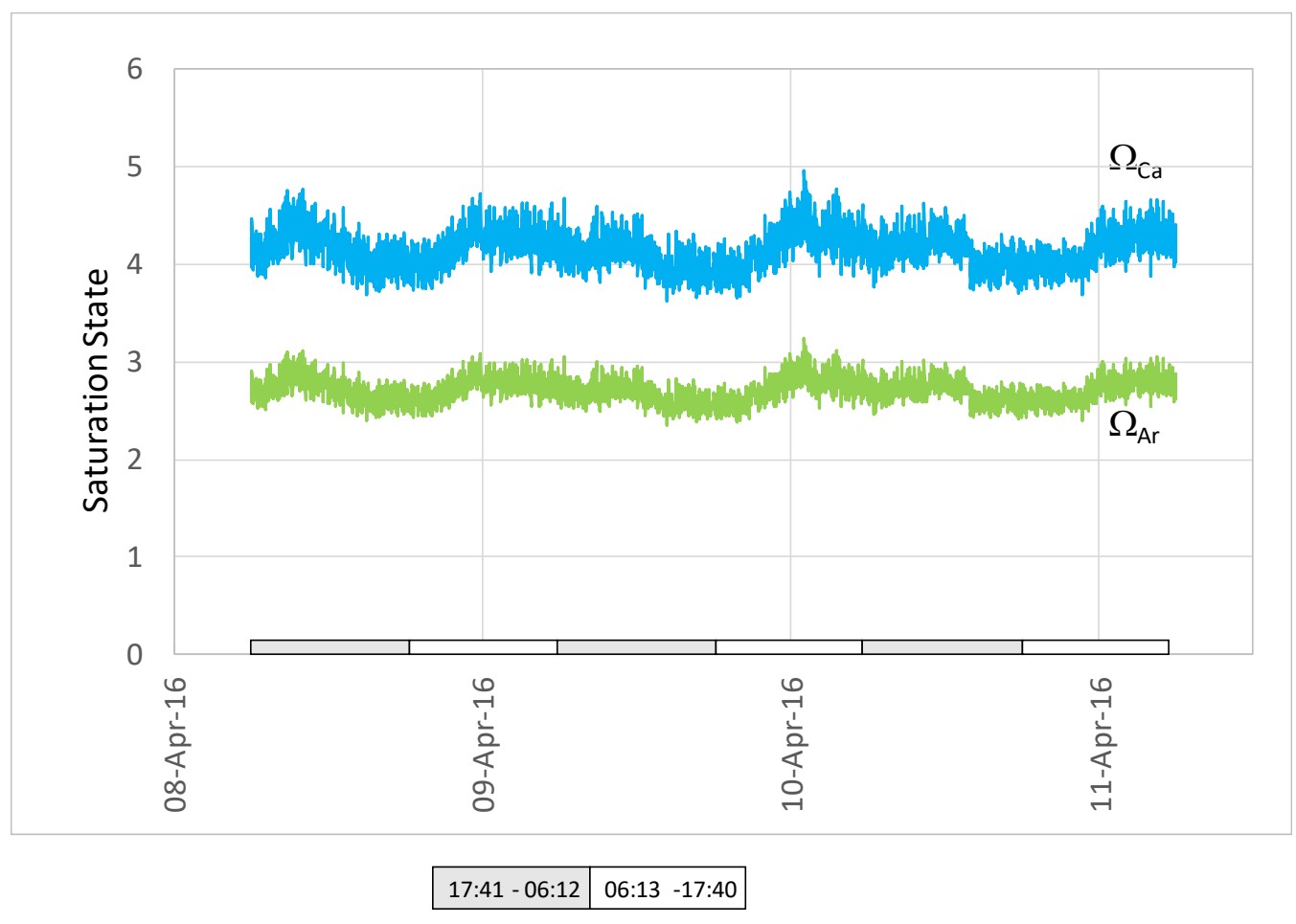

| 17:41 - 06:12 | 06:13 -17:40 |

**Figure 6. Aragonite and calcite saturation values calculated from data presented in Fig. 4.  Grey and white boxes indicate the time between sunset and sunrise.**

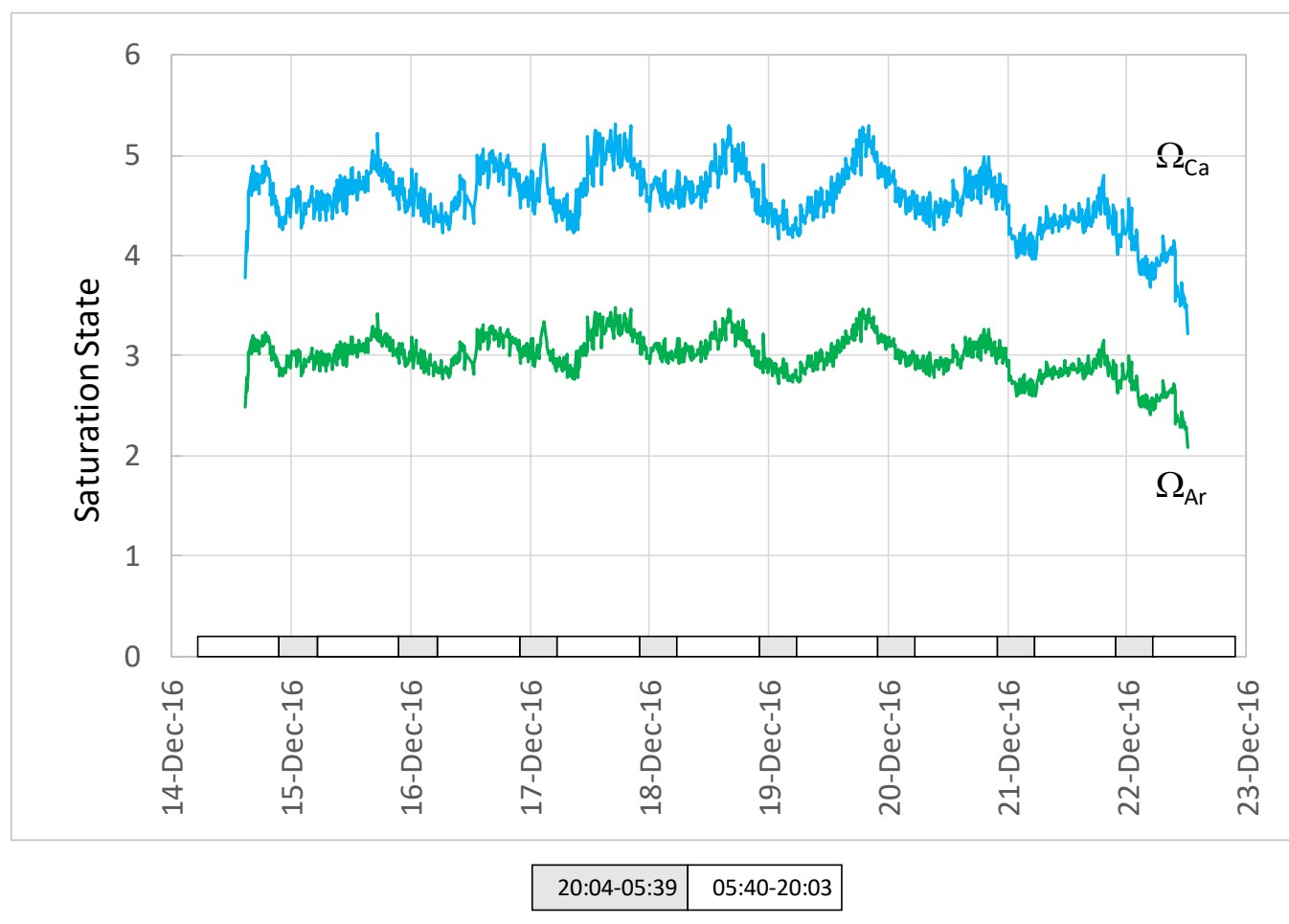


**Figure 7. Aragonite and calcite saturation values calculated from data presented in Fig. 5. Grey and white boxes indicate the time between sunset and sunrise.**

