# Peer review of "Continuous fluorescence-based monitoring of seawater pH in situ."

_Biogeosciences, 2017_

## Referee Comment (RC1) · Anonymous Referee #1 · 15 Aug 2017

General Comments

This manuscript describes the development and testing of a sensor for monitoring seawater pH, suitable for coastal and estuarine locations, and potentially deployable at depth. A relatively low cost, sensor that is simple to deploy and accurate and precise enough to detect changes on diurnal time scales has many potential applications. However, such a sensor needs to be very well characterised and its reliability assessed under a variety of conditions.

The sensor described in this paper potentially meets these criteria, however this is difficult to assess because not enough information is given. The presentation at The Oceans in a High CO2 World conference was good in that context, however more detail and more validation is required in a journal paper.

[Figure]

The paper is well laid out, and is very readable, though often the language is not precise – use of roughly, appearing to, approximately etc.

Specific comments

Line 21, spell [CO2] out in full the first time it is used.

Line 28, the potentiometric method for measurement of seawater pH can be accurate and precise if appropriate care is taken with temperature control, use of seawater buffers etc. An appropriate reference is required here, (Eg Dickson, A. G., Sabine, C. L., and Christian, J. R.: Guide to best practices for ocean CO2 measurements. PICES Special Publication 3. IOCCP Report No. 8, 191 pp, in: PICES Special Publication 3, 191 2007.)

The spectroscopic method should also be mentioned in this paragraph, as it is now a common method of measuring seawater pH, and indeed is used in the work described here in Lines 102 - 105. Relevant references should be included, such as the Dickson et al. Guide to Best Practices.

Line 48 – specify austral Autumn and Summer.

Line 49 – the alkalinity -salinity relationship determined by Lenton et al (2016) was shown to be valid at the open ocean and coastal IMOS sites. The relationship at the Chowder Bay site is likely to be different due to terrestrial influences , and this should be acknowledged.

Line 55 – repetitive wording needs editing.

Line 56 – specify Australia in the description of the location.

Line 90 – specify the brand of the "standard buffers", and what scale the pH is on (pHT, pHNBS or something else).

Line 91 – the C1 referring to the measured pH of the first buffer, can be confused with Cl (chloride) used earlier in the line. I suggest using a different symbol for the buffer

pH.

Line 98 – although you did not use tris buffer, because of its adverse effect on the electrode, it is necessary to tie your measurements back to the seawater scale, particularly as you state in line 96 that you are using the (total) pH scale. You should also mention the temperature of your calibration. Was it assumed that everything was at the stable room temperature of 22 oC (line 89), or did you measure the temperature?

Line 199 - how many calibration samples taken during the in situ deployment, and how many samples were taken? These should be noted on Figures 4 and 5.

Line 101 - specify the actions taken to minimise gas exchange during sampling.

Line 104 - more information here needed here on the validation against the CRMs – the precision and accuracy of your measurements is required.

Line 106 - replace the term "published" with "certified" .

Line110 - how were the corrections applied – an average offset, a drift, or some other method?

Lines 111 – 113, 117-119, and 124-126. These sentences are clumsy – separate into the ex situ and in situ methods.

Line 131 - specify what corrections were made to the raw data, and what scale the final pH is on (pHT?).

Line 138 - a reference is required for the influence of photodegradation on fluorophores.

Line 139 - How was the drift corrected for? Were the bottle samples used for this?

Line 146 - Use the symbol oC not the word degrees.

Line 157-157 Different symbols used for aragonite and calcite saturation, see Lines 132, 154-157, 181, 201 and Figs 6 and 7.

Line 163 - a reference is required for this sentence – what values of pH, S and T would

be expected?

Line 167 - what is the variation due to – is it degassing due to pumping, temperature changes or another reason?

Fig 1 - it would be useful to include an inset map of Australia with the location noted.

Fig 3 caption - It would be useful to specify in the caption that the red arrow show light conveyed to the spot is indicated by the red arrow, and light emitted from the spot is indicated by the blue arrows. Eg The insert shows light conveyed through the transparent vessel wall to the spot (red arrow) , and fluorescent light emitted from he spot back along the fibre (blue arrows).

Figs 4 and 5– Specify the pH scale, and mark the pH of the bottle samples on the plot.

Figs 6 and 7 captions, delete the sentence "Values below unity represent dissolution."

---

## Author Comment (AC1) · 24 Aug 2017

The authors wish to acknowledge the reviewers comments, and thank the reviewer for the detailed suggestions. We will take into consideration all these comments during our final review of the manuscript, and are confident that the improvements made will strengthen the work and make it a more valuable contribution. With our best regards, The authors.

---

## Referee Comment (RC2) · A. Dickson (Referee) · 25 Oct 2017

Although I was interested to see the application of the Presens pH-sensitive spots to looking at seawater pH variability, I am not comfortable recommending this manuscript for publication in its present form.

The title, and the abstract, indicate that this paper is to be considered as an assessment of using this technique to monitor estuarine pH, and hence (when used in conjunction with salinity information) to provide information about the time-dependent variation of the seawater $CO_2$ properties, and in particular the aragonite and calcite saturation states.

The authors however fail to make clear the likely overall uncertainty of their measure-

ments, a key parameter when assessing a measurement technique. Also, the data provided do not really cover a sufficient range of salinity to plausibly assess likely behavior in an estuary.

The calibration of the Presens sensor for use in seawater media seems odd, and is insufficiently described. In principle, a calibration curve was prepared documenting the instrument response as a function of the pH of a series of buffer solutions: phosphate buffers in a NaCl background (I = 0.7 M) . The pH values for the various buffer solutions -at a temperature of $\sim$22 °C - were themselves measured directly using a pH cell that had itself been calibrated against low ionic strength buffers ($\sim$0.1 M I suspect). The authors note that validation samples were collected and their pH measured spectrophotometrically (using m-cresol purple - though not purified), but the paper does not really show the direct comparison but simply comments that a "drift" in the fluorescence-based pH sensor was thus corrected for.

There is, I feel, so much wrong with this approach. First, it is not at all clear what really is being measured as pH (and this will cause a mismatch with the equilibrium constants used when the various saturation states are calculated), second I would have imagined that the response of the sensor dot would depend (at least to some extent) on temperature and salinity - this is not even mentioned.

Furthermore, the simple calibration of the conductivity sensor with a pure NaCl solution will likely affect the accuracy of estimated salinity values, and ultimately the accuracy of alkalinity values estimated from these.

Other notes (linked to line numbers)

103,4 The text refers to a commercial dye (Acros, and to the calibration for a purified dye. I would note that Acros was one of the worst commercial dyes, and that a recent paper in Marine Chemistry by Douglas & Byrne, suggests how to convert the behavior of the impure dye to approximate that of a pure dye.

[Figure]

123 It is stated that the conductivity probe was calibrated with a sodium chloride solution of known concentration. Surely one also needs a value for its conductivity, and a process for inferring seawater salinity from conductivity (rather than conductivity ratio - as implied by the definition of Practical Salinity).

133 Why "pCO2" here? It is not a measurement, nor is it discussed anywhere as a calculated value

133 The citation to "Dickson and Millero (1987) does not make clear which constants were chosen. The usual ones are those of Mehrbach, converted to the total hydrogen ion concentration pH scale, but a more explicit statement would be appropriate

136 The "precision" of the pH value is given as 0.022 - assuming this is some form of standard deviation, it is not clear how it was computed.

149 "psu" is not an appropriate unit abbreviation. The Practical Salinity Scale has the unit "1" I note in Figs. 4 and 5 that salinity is apparently in "ppt" - this may be a closer reflection of the calibration approach

155 two not "three" parameters?

156 The comment is made that the saturation state of aragonite has less "variability" than that of calcite. This is a necessary consequence of the definitions of saturation state whereby the ion product [Ca][CO3] is multiplied by 1/Ksp, and as the Ksp is different for aragonite and for calcite, so too is the multiplier with that for calcite being the larger.

182 The comment is made that temperature and salinity play an important role in seawater chemistry. This is, in part, because the various equilibrium constants are themselves function of T & S. But also, here alkalinity (the 2nd CO2 parameter required for calculations) is itself a function of salinity. - the m/s does not make this clear.

Figures I was surprised to see that the night/day cycle is identical in both April and December The time axis on the figures is hard to read (and needlessly varies from one

frame to another

---

## Author Comment (AC2) · 22 Nov 2017

**Response to referee (A. Dickson)**

> The authors thank Professor Dickson for his comments.  Please note responses are indented and follow the referee's comment.

Although I was interested to see the application of the Presens pH-sensitive spots to looking at seawater pH variability, I am not comfortable recommending this manuscript for publication in its present form.

The title, and the abstract, indicate that this paper is to be considered as an assessment of using this technique to monitor estuarine pH, and hence (when used in conjunction with salinity information) to provide information about the time-dependent variation of the seawater CO2 properties, and in particular the aragonite and calcite saturation states. The authors however fail to make clear the likely overall uncertainty of their measurements, a key parameter when assessing a measurement technique. Also, the data provided do not really cover a sufficient range of salinity to plausibly assess likely behaviour in an estuary.

> The intent of the work was to determine the utility of the Presens sensor for use in shallow inshore marine systems to detect variability in seawater carbonate chemistry over timescales of minutes to hours.  While accuracy was an important factor, our intention, with proof-of-concept data, was to demonstrate that this technique is capable of showing clear patterns of variability.

> To avoid confusion, we propose changing the title to:
> "Continuous fluorescence-based monitoring of seawater pH *in situ*."  This excludes the word estuary and will hopefully avoid any implication that the research is generally applicable to estuaries. While the study location can be said to be in the estuary, in reality it is close to the ocean and is primarily influenced by the ocean.

> The abstract is also modified to now include an estimate of uncertainty.
> Line 13: "…background variability.  While the stated phase angle resolution of the lifetimes fluorometer translated into pH units was  +/- 0.0028 pH units, the precision of calculated pH was +/- 0.022, indicating error associated with calibrating the device against known reference samples.  Diel variability…"

> Further:
> Line 45: "…in the setting of an ocean-water dominated estuarine environment."
> Line 52: "… in surface waters.  The study focussed on natural fluctuations in seawater pH and the capability of the fluorescence technique to address these fluctuations. Determining the accuracy of pH values obtaining from the fluorescence device was not an objective, as correct determinations would require considerable resources that were not available at the time of study."

The calibration of the Presens sensor for use in seawater media seems odd, and is insufficiently described. In principle, a calibration curve was prepared documenting the instrument response as a function of the pH of a series of buffer solutions: phosphate

buffers in a NaCl background (I = 0.7 M) . The pH values for the various buffer solutions -at a temperature of _22 _C - were themselves measured directly using a pH cell that had itself been calibrated against low ionic strength buffers (_0.1 M I suspect). The authors note that validation samples were collected and their pH measured spectrophotometrically (using m-cresol purple - though not purified), but the paper does not really show the direct comparison but simply comments that a "drift" in the fluorescence based pH sensor was thus corrected for.

> We now show a direct comparison of data obtained using the spectrophotometric technique and the fluorometric technique as indicated by the data points in Figures 4 and 5.

There is, I feel, so much wrong with this approach. First, it is not at all clear what really is being measured as pH (and this will cause a mismatch with the equilibrium constants used when the various saturation states are calculated),

> Our initial measurement of pH was conducted using the potentiometric technique, where we calibrated the glass electrode using standard buffers to ensure the electrode provided a reasonable estimate of pH, notwithstanding any effect of increased salinity on subsequent measurements.  The phosphate buffers then enabled us to characterise the relationship across a range of pH values (with salinities similar to seawater) and various values of fluorescence lifetime.  From this relationship we could then interpolate and use the fluorescence lifetime to indicate the "pH".  However, only when the fluorescence sensor was directly compared with the spectrophotometric measurements of a collected sample were we able to apply corrections to the fluorescence-derived values for pH, and refer to them as "$pH_T$" in line with the spectrophotometric $pH_T$ values.

> We also discuss the use of constants when calculating the seawater saturation state

 second I would
have imagined that the response of the sensor dot would depend (at least to some extent) on temperature and salinity - this is not even mentioned.

> This will be  mentioned in the introduction.
> L 44 "…changes in fluorescence intensity.  As the fluorophores used in this study are sensitive to changes in temperature and salinity, careful measurement of these parameters remains essential.".

Furthermore, the simple calibration of the conductivity sensor with a pure NaCl solution will likely affect the accuracy of estimated salinity values, and ultimately the accuracy of alkalinity values estimated from these.

> We agree that the use of NaCl to calibrate the salinity sensor is not ideal.  However the aim of the exercise was to demonstrate the ability of the fluorescence-based technique to show diel variability in seawater carbonate chemistry.

Other notes (linked to line numbers)
103,4 The text refers to a commercial dye (Acros, and to the calibration for a purified dye. I would note that Acros was one of the worst commercial dyes, and that a recent paper in Marine Chemistry by Douglas & Byrne, suggests how to convert the behaviour of the impure dye to approximate that of a pure dye.

> In the introduction we now make it clear that our rationale was to demonstrate that the fluorescence method that can show natural variability of pH within an estuary over time scales of minutes to hours rather than provide highly accurate pH data.

> Line 52: "… in surface waters. The study has focussed on natural fluctuations in seawater pH and the capability of the fluorescence technique to address these fluctuations. Determining the accuracy of pH values obtained from the fluorescence device to the third decimal place was not an objective.

> While we did not correct for dye impurities - we agree that these new papers are important to cite.

> Line 104: "and calculations from Liu et al. (2011). No attempt was made to correct for dye impurities (Douglas and Byrne 2017a) nor variable salinity and temperature (Douglas and Byrne 2017b), where uncertainty attributable to these combined factors is within approximately ± 0.004 (Douglas and Byrne 2017b)."

123 It is stated that the conductivity probe was calibrated with a sodium chloride solution of known concentration. Surely one also needs a value for its conductivity, and a process for inferring seawater salinity from conductivity (rather than conductivity ratio - as implied by the definition of Practical Salinity).

> A value for salinity in psu was sufficient to calculate seawater carbonate chemistry, and no attempt was made to infer conductivity from the sensor.

> Line 123: The sensor was calibrated directly against a pure NaCl solution across a range of concentrations (0 to 40 ppt) at temperatures similar to that expected of seawater at the sampling location (23°C).

133 Why "pCO2" here? It is not a measurement, nor is it discussed anywhere as a calculated value

> Alll reference to pCO2 has been removed from the text.

133 The citation to "Dickson and Millero (1987) does not make clear which constants were chosen. The usual ones are those of Mehrbach, converted to the total hydrogen ion concentration pH scale, but a more explicit statement would be appropriate

> Line 132: "Corrections were made to the raw data if required, and final pH, pCO2, temperature and alkalinity values were used (with equilibrium constants from

Mehrbach et al., 1973 refit by Dickson and Millero, 1987) to calculate $\Omega_{Ca}$ and $\Omega_{Ar}$ in CO2Sys (V2.1)."

136 The "precision" of the pH value is given as 0.022 - assuming this is some form of standard deviation, it is not clear how it was computed.

Suitable text describing the method for determining the precision of measurements has been added to the methods section:

Line 108: An additional series of continuous measurements were made in a static constant-temperature 20 L seawater system over an approximately three-hour period to determine the variability in measured pH.  The resolution attributable to the lifetimes decay fluorometer (as stated by the manufacturer) can be matched with corresponding values of pH to provide resolution of pH measurement at the instrument level.

Line 136: "The precision of calculated $pH_T$ (defined as the standard deviation of three hours of consecutive measurements of a static seawater solution) was ± 0.022 pH units. The phase angle resolution of the lifetimes fluorometer was stated by the manufacturer as 0.05 degrees, and this translated to an instrument resolution of 0.0028 pH units.

149 "psu" is not an appropriate unit abbreviation. The Practical Salinity Scale has the unit "1" I note in Figs. 4 and 5 that salinity is apparently in "ppt" - this may be a closer reflection of the calibration approach

Line 149: "and ranged from 33.0 to 34.3 ppt…"

155 two not "three" parameters?

Line 155: "smoothed approximation of diel variation in these two parameters (Fig. 6).

156 The comment is made that the saturation state of aragonite has less "variability" than that of calcite. This is a necessary consequence of the definitions of saturation state whereby the ion product [Ca][CO3] is multiplied by 1/Ksp, and as the Ksp is different for aragonite and for calcite, so too is the multiplier with that for calcite being the larger.

Thankyou for your insight
Line 156 "As a consequence of the definitions of saturation state, and form-specific values of $K_{sp}$ for aragonite and calcite, $\Omega_{ara}$ was slightly less variable than $\Omega_{cal}$ (with values between 2.5 and 3.1, or 3.8 to 5.4 units respectively, Fig. 6).

182 The comment is made that temperature and salinity play an important role in seawater

chemistry. This is, in part, because the various equilibrium constants are themselves function of T & S. But also, here alkalinity (the 2nd CO2 parameter required for calculations) is itself a function of salinity. - the m/s does not make this clear.

> Line 182: "These data show the important role that temperature and salinity play in determining seawater carbonate chemistry (Millero 1995).  Further, as the saturation state in this study is in part derived from alkalinity which is in turn derived from salinity and temperature, the importance of obtaining accurate measurements of temperature and salinity is to be emphasised.

Figures I was surprised to see that the night/day cycle is identical in both April and December The time axis on the figures is hard to read (and needlessly varies from one frame to another

> Good spotting - we corrected this error in Figures 4 to 7 and  the time axis is larger and now uniform across all figures.

> Additional references to add (Douglas and Byrne 2017a, and 2017b, Dickson 2010 [in Guide to best practises… edited by Riebesell et al])

---

## Author Comment (AC3) · 22 Nov 2017

**Response to referee #1**

The authors thank the anonymous referee for their comments. Our responses are follow the referee's comments in the indented text. Corrected text is in quotes.

The paper is well laid out, and is very readable, though often the language is not precise – use of roughly, appearing to, approximately etc.

The manuscript was checked for imprecise language and corrected throughout.

Specific comments
Line 21, spell [CO2] out in full the first time it is used.

Line 21 "…in response to elevated concentration of atmospheric carbon dioxide ($CO_2$), with a decline in pH and an increase in the partial pressure of dissolved $CO_2$ ($pCO_2$) over the coming decades."

Line 28, the potentiometric method for measurement of seawater pH can be accurate and precise if appropriate care is taken with temperature control, use of seawater buffers etc. An appropriate reference is required here, (Eg Dickson, A. G., Sabine, C. L., and Christian, J. R.: Guide to best practices for ocean CO2 measurements. PICES Special Publication 3. IOCCP Report No. 8, 191 pp, in: PICES Special Publication 3, 191 2007.)

Line 28 "Seawater pH has been commonly measured with a potentiometric technique using glass electrodes. While this method can be accurate and precise if appropriate care is taken with temperature control and the use of seawater buffers (Dickson, 2010), this technology suffers from a gradual…".

The spectroscopic method should also be mentioned in this paragraph, as it is now a common method of measuring seawater pH, and indeed is used in the work described here in Lines 102 - 105. Relevant references should be included, such as the Dickson et al. Guide to Best Practices.

Line 32. "… pers.comm). Spectroscopic techniques for measuring seawater pH use a pH-sensitive dye that assumes different absorbance spectra depending on pH. While the method can provide pH estimates with an uncertainty less than 0.01, variability in the quality of the dye obtained from commercial suppliers can cause the true extinction coefficients associated with a particular dye to differ slightly from published extinction coefficient values. Consequently, uncertainty associated with this technique is generally assumed to be about 0.01 pH units (Dickson, in Riebesell et al. 2010)

Reference is "Dickson, A.G. "The carbon dioxide system in seawater: equilibrium chemistry and measurements". *In* Riebesell, V.J., Fabry, V.J., Hansson, L. and Gattuso, J.-P. 2010. Guide to best practises for ocean acidification research and data reporting. Luxembourg, Publications Office of the European Union."

Line 48 – specify austral Autumn and Summer.

"… two multi-day intervals in austral Autumn and Summer"

Line 49 – the alkalinity -salinity relationship determined by Lenton et al (2016) was shown to be valid at the open ocean and coastal IMOS sites. The relationship at the Chowder Bay site is likely to be different due to terrestrial influences, and this should be acknowledged.

"Electrical conductivity and temperature were also measured, and an approximation for alkalinity was derived from a salinity-alkalinity relationship reported for the Australasian region (Lenton et al. 2016).  We acknowledge that the relationship between salinity and alkalinity is subject to terrestrial influence, and consequently our estimate of alkalinity remains an approximation."

Line 55 – repetitive wording needs editing.

"Real-time measurements were made of seawater in the Sydney Harbour…"

Line 56 – specify Australia in the description of the location.

"Chowder Bay, NSW, Australia (-33…"

Line 90 – specify the brand of the "standard buffers", and what scale the pH is on ($pH_T$, $pH_{NBS}$ or something else).

"was calibrated against standard buffers (pH 7.02 and 10.06 at 20°C, Amalgamated Instruments Co. Pty Ltd, Australia)

Line 91 – the C1 referring to the measured pH of the first buffer, can be confused with Cl (chloride) used earlier in the line. I suggest using a different symbol for the buffer pH.

"isotonic to ~34.5 ppt NaCl ($I$ = 0.7 M) with B1=8.33 pH and B2=4.00 pH (where B1 and B2 are the measured pH of each of the buffers).

Line 98 – although you did not use tris buffer, because of its adverse effect on the electrode, it is necessary to tie your measurements back to the seawater scale, particularly as you state in line 96 that you are using the (total) pH scale. You should also mention the temperature of your calibration. Was it assumed that everything was at the stable room temperature of 22 oC (line 89), or did you measure the temperature?

We used the total pH scale as recommendation by Dickson (in Riebesell et al. 2010), who also advised against the use of the seawater pH scale for seawater pH measurements.  We selected the total scale when calculating omega values, to ensuret hat the equilibrium constants used to determine seawater carbon chemistry values were consistent with the scale we used to measure and report pH.  The room

temperature of 22°C was stable during the calibration procedure, and buffer temperature was 22 ± 0.5 °C during the calibration procedure.

Line 98: "We did not use Tris as a buffer as it is known to poison single junction Ag/AgCl electrodes, and report pH in the total hydrogen ion scale (as recommended by Dickson (in Riebesell et al. 2010). Both room and buffer temperature were stable during the calibration procedure at 22°C and 22 ± 0.5 °C, respectively."

Line 100 - how many calibration samples taken during the in situ deployment, and how many samples were taken? These should be noted on Figures 4 and 5.

Line 100: "At the beginning and end of the *in situ* deployment, water samples were collected adjacent to the submerged pH monitor and transported to the nearby lab at SIMS. At each sampling time, three 200 mL glass jars were completely filled, and lids secured underwater to eliminate air bubbles. This process was repeated at least three times, with approximately 30 minutes between each sampling time.

This information will be added to Figs 4 and 5

Line 101 - specify the actions taken to minimise gas exchange during sampling.

See above

Line 104 - more information here needed here on the validation against the CRMs – the precision and accuracy of your measurements is required.

Line 104: "Spectrophotometric measurements of $pH_T$ were validated using seawater certified reference material (CRM, Dickson Standard Batch 145). The precision of the spectrophotometric technique in our laboratory is typically ±0.006 at temperatures between 17 to 24°C (standard deviation), with an accuracy very much dependent on the dye quality. We did not apply corrections for dye impurities, temperature and salinity variations as discussed in Douglas and Byrne (2017a,b) because obtaining such highly accurate pH values was not an objective of this study."

Line 106 - replace the term "published" with "certified" .

Line 106: "the CRM using published values (derived from certified standards) for total alkalinity ($A_T$), dissolved inorganic carbon (DIC) and salinity at the same"

Line110 - how were the corrections applied – an average offset, a drift, or some other method?

Line 110: "Corrections were made against spectrophotometric measurements of the reference samples by calculating two average offsets between values of collected samples measured using the fluorometric device and the spectrophotometric device.

One offset was derived for the beginning of the deployment and one for the end. The gradual decline in measured values over time was corrected assuming a constant linear drift."

Lines 111 – 113, 117-119, and 124-126. These sentences are clumsy – separate into the ex situ and in situ methods.

The sentences have been corrected.

Lines 111-113: "*In situ* measurements were taken ten times per minute and averaged over one minute before analysis. *Ex situ* measurements were taken once per minute and a running average of 10 minutes of the *ex situ* measurements is presented."

Lines 117-119: "*In situ* measurements were taken ten times per minute and averaged over one minute before analysis. *Ex situ* measurements were taken twice per minute and a running average of 10 minutes of the *ex situ* measurements is presented."

Lines 124-126: "*In situ* measurements were taken ten times per minute and averaged over one minute before analysis. *Ex situ* measurements were taken three times per minute and a running average of 10 minutes of the *ex situ* measurements is presented.

Line 131 - specify what corrections were made to the raw data, and what scale the final pH is on (pHT?).

Line 131: "Drift and offset corrections were made to the raw data as required (described above), and final pH, salinity, temperature and alkalinity values were used…" to calculate $\Omega_{Ca}$ and $\Omega_{Ar}$ in CO2Sys (V2.1) (Dickson and Millero 1987). The final pH is reported on the total scale."

Line 138 - a reference is required for the influence of photodegradation on fluorophores.

A references is added to Line 138:
"Gradual bleaching is known to influence fluorophores (Lakowicz 2006), contributing to a steady drift."
Reference: Lakowicz J.R. 2006. Principles of Fluorescence Spectroscopy". 3[rd] edition. Springer.

Line 139 - How was the drift corrected for? Were the bottle samples used for this?

Line 139: see above

Line 146 - Use the symbol oC not the word degrees.

Line 146: "…temperature varied between 1 and 2°C,

Line 157-157 Different symbols used for aragonite and calcite saturation, see Lines 132, 154-157, 181, 201 and Figs 6 and 7.

Consistent use of the term $\omega_{Ca}$ and $\omega_{Ar}$ throughout text and Figs 6 and 7

Line 163 - a reference is required for this sentence – what values of pH, S and T would be expected?

> Line 163: "  would expect for the harbour at that time of year (e.g. pH ~8.0, S=~35 ppt, T= ~18°C, pers.obs.)

Line 162: "Values of pH, salinity and temperature were all within what one would expect for the harbour at that time of year."

> See above

Line 167 - what is the variation due to – is it degassing due to pumping, temperature changes or another reason?

> Line 167: "However, the dissolved gas content of pumped water can vary relative to the source water due to degassing and temperature changes, making measurements potentially less representative of the water to be sampled"

Fig 1 - it would be useful to include an inset map of Australia with the location noted.

[Figure]

Fig 3 caption - It would be useful to specify in the caption that the red arrow show light conveyed to the spot is indicated by the red arrow, and light emitted from the spot is indicated by the blue arrows. Eg The insert shows light conveyed through the transparent vessel wall to the spot (red arrow) , and fluorescent light emitted from he spot back along the fibre (blue arrows).

Figure 3. Schematic of pH monitoring system with EOM-pH-mini (left), and fibre optic conveying light to fluorescent spot attached to inside of vessel (right). The insert shows light conveyed through the transparent vessel wall to the spot (red arrow), and fluorescent light emitted from the spot back along the fibre (blue arrows).

Figs 4 and 5– Specify the pH scale, and mark the pH of the bottle samples on the plot.

Ok this will be done.

Figs 6 and 7 captions, delete the sentence "Values below unity represent dissolution."

Ok this will be done.

---

## Author Response (AR3)

**Continuous fluorescence-based monitoring of seawater pH *in situ*.**

**Authors responses to reviewers**

Thankyou to both the associate editor and reviewer 2.  Many of the minor text changes recommended by reviewer 2 can be found directly in the document.  Additional comments follow according to line numbers on the marked up version of the document (not the resubmission):

Line 12: replaced with " The fluctuations observed in pH over intervals of minutes to hours could be distinguished from background noise"
14: plus/minus removed
14: replaced with "While the stated phase angle resolution of the lifetimes fluorometer translated into pH units was $\pm$ 0.0028 pH units, the repeatability standard deviation of calculated pH was  0.007 to 0.009"
31: replaced with "gradual drift that may be attributable to changes in the strain-induced asymmetry potential of the glass bulb and reference junction effects"
31: replaced with In addition, common reference  electrode designs
33: replaced with "incorporation of double (or even quadruple) junction salt bridges"
37 true replaced with "apparent"
39: replaced with "about 0.015 pH units (Douglas and Byrne 2017b)."
44: sentence replaced with "However this introduces problems in locations where chloride concentrations may rapidly fluctuate, such as some euryhaline estuarine environments"
50: "s" removed from lifetimes
63: replaced with "observe"
64: replaced with "uncertainty"
64: replaced with "precise and accurate pH determinations"
104: replaced with "A pH cell with an Ag/AgCl reference electrode"
108: replaced with "the pH cell"
112: the graph has less than 10 data points and we believe would not add much useful information to the manuscript
124: replaced with "repeatability"
124: Plus/minus removed
128: replaced with 0.015
128: replaced with "The measured $pH_T$ was corrected according to the calculated $pH_T$ of the CRM using published values for total alkalinity ($A_T$), dissolved inorganic carbon (DIC) and salinity at the same temperature.  This correction eliminates much of the variability due to dye quality, light source and cuvette scratches."
150: added "Uncertainty due to differences in conductivity of a pure NaCl solution at 35 ppt and S=35 was assumed to be small.  "
161: the drift correction was also made with the April data, however the figure shows a subset of the data to better illustrate a diel pattern and the variability inherent in the data, and does not show the comparisons which were made at the beginning and end of the deployment.  See also modified text at line 131" A subset of consecutive pH measurements over a 100 minute period was sampled during each of the two deployments to determine the variability in measured pH as repeatability (standard deviation).  The resolution attributable to the lifetime decay fluorometer…"

165: This is corrected so that the resolution of the fluorescence spot as stated by the manufacturer is now stated as 0.01 pH units

169: gradual photobleaching could be minimised by ensuring the sensor spots were in complete darkness.

Paragraph: "Seawater collected and pumped to the sensor varied by 0.08 pH units from 7.97 to 8.05 over the three-day interval in Autumn (Fig. 4). The phase angle resolution of the lifetimes fluorometer was stated by the manufacturer as 0.05 degres, and this translated to an instrument resolution of 0.0028 pH units when calibrated against the phosphate buffer mixtures. The measurement technique is also constrained by the stated resolution of 0.01 pH units) of the fluorescence spots. The repeatability standard deviation of calculated $pH_T$ (derived from 100 minutes of consecutive measurements during each deployment was 0.009 and 0.007 pH units for the April and December deployments respectively. The best precision can be achieved by averaging multiple measurements, although this will of course hide real variability. Gradual bleaching is known to influence fluorophores (Lakowicz 2006), contributing to a steady drift. This was accounted for with corrections based on samples collected *in situ* as described above. In this case, the decline in pH was 0.0136 pH units per day which is approximately three-fold the drift associated with measurement-induced bleaching of 0.003 pH units per 1000 measuring points as stated by the manufacturer. pH measured over summer ranged from 7.95 to almost 8.15 units, some 0.1 units more basic than Autumn values (Fig. 5)."

188: redundant part removed, and sentence modified "As a consequence of the definition of saturation state and due to the higher solubility of aragonite, and from specific values of $K_{sp}$ for aragonite and calcite, $\Omega_{Ar}$ was slightly less variable than $\Omega_{Ca}$ (with values between 2.5 and 3.1, or 3.8 to 5.4 units respectively, Fig. 6)."

198: additional material added to explain photobleaching drift: "...marine surface waters. While an attempt was made to minimise exposure of the sensor spot to ambient light, the rate of drift observed in the instrument due to photobleaching may have been greater during the day. Photobleaching induced drift could be further reduced by excluding all ambient light and by reducing the sampling frequency. This would be needed for longer term deployments. Calibration against high accuracy pH measurements at least at the beginning and end of a deployment, and more frequently for an extended deployment, would be required to maintain suitably accurate data."

211: "A metal housing with regular bulkheads along its length would increase pressure resistance and provide an instrument capable of withstanding deep ocean pressures, although a system with improved precision would be required for deep ocean work as the variation in pH at depth is slight relative to surface waters. "

236: replace 0.022 with 0.007 pH units

315. (Figure 4) No colourimeteric measurements are shown in Figure 5 as this is a three day representation of a longer time series, where the colourimetric measurements and drift corrections were made at the beginning and end of the deployment.

[revised manuscript text omitted]